# Disclosing an In-Frame Deletion of the Titin Gene as the Possible Predisposing Factor of Anthracycline-Induced Cardiomyopathy: A Case Report

**DOI:** 10.3390/ijms23169261

**Published:** 2022-08-17

**Authors:** Yu-Wei Chang, Hui-Ying Weng, Shih-Feng Tsai, Frank Sheng Fan

**Affiliations:** 1Department of Laboratory, Taitung Hospital, Ministry of Health and Welfare, Taitung 95043, Taiwan; 2Department of Nursing, Meiho University, Pingtung 91202, Taiwan; 3Biomedical Industry Ph.D. Program, National Yang Ming Chiao Tung University, Taipei 112304, Taiwan; 4Institute of Molecular and Genomic Medicine, National Health Research Institute, Miaoli 350, Taiwan; 5Department of Medicine, Taitung Hospital, Ministry of Health and Welfare, Taitung 95043, Taiwan

**Keywords:** anthracycline-induced cardiotoxicity, chemotherapy-related cardiomyopathy, titin

## Abstract

Anthracycline-induced cardiomyopathy has been noted as a non-neglectable issue in the field of clinical oncology. Remarkable progress has been achieved in searching for inherited susceptible genetic deficits underlying anthracycline cardiotoxicity in the past several years. In this case report, we present the preliminary results of a genetic study in a young male patient who was treated with standard dose anthracycline-based chemotherapy for his acute myeloid leukemia and attacked by acute congestive heart failure after just two courses of therapy. After a survey of 76 target genes, an in-frame deletion of the titin gene was recognized as the most possible genetic defect responsible for his cardiomyopathy caused by anthracycline. This defect proved to pass down from the patient′s mother and did not exist in seven unrelated chemotherapy-treated cancer patients without chemotherapy-induced cardiomyopathy and four other healthy volunteer DNA donors.

## 1. Introduction

Tremendous development in the field of anticancer therapy has led to a remarkably improved prognosis of cancer patients worldwide in the past three decades. Nevertheless, those life-saving therapeutic agents are not without adverse effects. Among them, cardiovascular toxicity becomes a peculiar concern with a variety of emergent presentations including myocardial ischemia, hypertension, cardiomyopathy, chest pain, pericarditis, hypotension, and arrythmia [1,2]. Chemotherapy-related cardiomyopathy (CCM) is currently classified into two types: type one, the permanent damage type, is mainly caused by chemotherapeutic drugs such as doxorubicin, daunorubicin, epirubicin, idarubicin, mitoxantrone, and cyclophosphamide, while type two, the reversible damage type, is strongly associated with monoclonal antibody trastuzumab and protein kinase inhibitors such as sunitinib and lapatinib [3]. Most of the cardiotoxic type one agents belong to the anthracycline group, which has a lifetime cumulative-dose relationship with the risk of cardiac failure. The mechanism of anthracycline-induced cardiotoxicity is proposed to be DNA double-strand break, the decreased expression of antioxidative enzymes, and impaired electron transport chains resulting from anthracycline binding with topoisomerase 2β, leading to the death of cardiac myocytes, production of reactive oxygen species, and mitochondrial dysfunction [3,4].

Although risk factors of anthracycline-induced cardiotoxicity have been demonstrated to include cumulative dose, very young or old age, female sex, prior cardiac diseases, existing cardiovascular threatening items, and concomitant treatment with trastuzumab, recent studies revealed an association between underlying genetic variants and anthracycline cardiotoxicity [5]. Due to the limited efficacy of present cardioprotective agents, for example, dexrazoxane, angiotensin-converting enzyme inhibitors, angiotensin II receptor blockers, and beta-blockers [6], prophylactic strategies for anthracycline cardiotoxicity have appealed to the use of prediction models and the precedent screening of associated pharmacogenetic variants for cancer patients who are to receive anthracycline therapy [7,8]. Herein, we present a young man with severe heart failure after taking just two courses of anthracycline-based chemotherapy for his acute myeloid leukemia and report the preliminary results of investigating possible underlying predisposing defects in his genome.

## 2. Materials and Methods

Next-generation sequencing (NGS) and data analysis.

The patient accepted our proposal that a survey of his genome for detecting probable genetic variants related to anthracycline cardiomyopathy might contribute a lot to medicine. After he signed his informed consent and the project was approved by our cooperative Institutional Review Board (IRB number: TYGH109047), his peripheral blood mononuclear cells (PBMCs) were sampled for genomic DNA extraction. Subsequently, NGS was used to establish the whole exome sequences (WES) by Genomics BioSci & Tech. (New Taipei City, Taiwan). The methodology was based on Agilent SureSelect system exome capture and sequencing protocol. The vcf files of WES data archived from Genomics Inc. were uploaded to QIAGEN Clinical Insight Interpret (QCII, QCI^TM^, QIAGEN, Hilden, Germany) [9]. After reading and aligning the WES data to human reference genome sequence (hg19), QCII software automatically calls sequence variants and generates a computed report based on the population frequency filter (excluding >1% gnomAD in the East Asian population). To interpret the relationship between the gene variants and pathogenic consequences, we used the specific terms based on the guidance of the American College of Medical Genetics and Genomics (ACMG), including “pathogenic”, “likely pathogenic”, “uncertain significance”, “likely benign”, and “benign” to describe the phenotypes which may be caused by target gene variants [10]. For the data visualization approach, Integrative Genomics Viewer (IGV) was used to generate the genomic mapping graph [11].

We focused on 76 target genes for a comparison analysis, according to the previously published literature. These targets cover most of the genes in which meaningful variants had been frequently detected in patients or experimental animal models with anthracycline-related cardiotoxicity [12,13,14,15]. A total of 76 target genes are listed in Table 1.

## 3. Case Presentation

A 20-year-old man was diagnosed as having acute myeloid leukemia with chromosome changes as del (9) (q22) and +10 at a medical center in Taipei city in February 2017. He was treated with two consecutive courses of chemotherapy with standard doses of cytarabine (200 mg/m^2^/day for 7 days) plus daunorubicin (45 mg/m^2^/day for 3 days). Complete remission was achieved but, unfortunately, severe congestive heart failure developed soon after the second course of chemotherapy. His left ventricular ejection fraction fell to 11% initially and recovered to 34% after intensive cardiologic care using a combination regimen composed of bisoprolol, valsartan/sacubitril, ivabradine, and spironolactone. Like many other leukemia and cancer patients in Taiwan, he did not take upfront dexrazoxane as a cardioprotective agent along with systemic chemotherapy. Although he had not received further therapy for his leukemia, the disease did not relapse when he attended our hospital for a routine medical follow-up study in February 2020. At that time, an echocardiogram showed that he had a dilated left ventricular chamber with an ejection fraction calculated at around 45%, showing much improvement from the previous estimation. His electrocardiogram showed a normal sinus rhythm without abnormal patterns. The hemoglobin level (15.5 g/dL) and platelet count (151,000/µL) were within reference ranges. There was a mild leukopenia (3600/µL) with an adequate differential distribution (neutrophil 66.4%, lymphocyte 22.1%, monocyte 8.7%, eosinophil 2.5%, and basophil 0.3%).

Based on the WES analysis with QCII, a preliminary data analysis on this case revealed a total of 319 variants related to the 76 target genes described above. After excluding >1% gnomAD in the East Asian population (benign polymorphism) through the population frequency filter provided by QCII, we concluded that three individual gene mutations, *TTN*, *MYO1A*, and *RYR2*, were found to be worth investigation (Table 2). A further study obtained from the QCII computed database indicated that only *TTN* variants contributed abnormal phenotypes—*YO1A* and *RYR2* were classified to be normal functional variants. Despite this, a total of 60 variation sites were detected on the *TTN* gene which encodes titin, a major component of sarcomere in cardiac muscle. Among them, an in-frame deletion, *TTN*: c.55637_55639delAAG, which led to the loss of a glutamic acid, turned out to be the most likely meaningful *TTN* gene variation of our CCM index patient (Figure 1). Furthermore, the QCII evidence base referred the Human Gene Mutation Database (HGMD^®^) and presented insight findings, depicting *TTN* variants significantly associated with cardiomyopathy; for example, the *TTN* missense mutation (c.7060C > T) with the R2354C substitution was validated as the pathogenic variant in hypertrophic cardiomyopathy [16]. Moreover, as reported in a cohort study including pathogenic variants and variants of unknown significance (VUS), increased variant burden was reported to be associated with dilated cardiomyopathy (DCM) [17]. Altogether, although the novel *TTN* variant was classified to be of uncertain significance in pathogenicity, the QCII knowledge base contains strong evidence to compute and predict the association between this *TTN* deleterious variation and hypertrophic cardiomyopathy.

The WES data were read and aligned to the hg19 reference genome. The visualization view of genomic mapping showed that AAG in-frame deletion occurred in the *TTN* gene of CCM case (TAIT-CCM-00), which caused the non-translation of glutamic acid (E) and, finally, altered the amino acid sequence.

The patient′s parents and elder brother, seven unrelated cancer patients (colon cancer—five, breast cancer—one, aggressive lymphoma—one), and four healthy volunteers were included in the extended phase of study after they gave their written informed consent (Table 3). No cardiomyopathy developed in any of the cancer patients who had all taken systemic chemotherapy without concurrent cardioprotectant dexrazoxane for their diseases. Of note, when comparing to the CCM patient with *TTN* in-frame deletion, we found that the lymphoma case (TAIT-CCM-13, see Table 4) with normal functional *TTN* alteration had no significant cardiomyopathy after receiving anthracycline therapy. Furthermore, some *TTN* loss function variants detected in different cancers remained low risk for cardiomyopathy after non-anthracycline chemotherapy in the analysis. WES was established from their genomic DNA and went through the same analytic process as described above, with one colon cancer patient′s DNA sample failing to pass quality control (unique patient number TAIT-CCM-08) and thirteen WES sets, in addition to the patient′s completed the analysis. The final results do suggest that the in-frame deletion *TTN* variant (*TTN*: c.55637_55639delAAG) with an alternative phenotype might be a highly possible causing factor for anthracycline-derived CCM.

The in-frame deletion *TTN*: c.55637_55639delAAG detected in the patient was also disclosed to exist in his mother and elder brother, thus, proving it to be a maternal-side inheritance. This mutation of interest could not be found in the patient′s father and all the other unrelated DNA donors. Nevertheless, few variant alternative *TTN* mutations identified to be of uncertain significance or likely benign in the QCII analysis were revealed in different DNA donors without cardiomyopathy besides the index patient (Table 4).

## 4. Discussion

Recent progress in the molecular pathophysiology study of hypertrophic and dilated cardiomyopathy has found meaningful deficits of genes involved in sarcomere composition and function [18]. Genetic deficits in hypertrophic cardiomyopathy result in important conformation changes which interfere with relaxation and energy preservation, leading to dilated cardiomyopathy. It has been revealed that a prevalence of truncating variants of titin, encoded by *TTN*, leads to a contractile dysfunction of sarcomere [19]. A survey of underlying genetic variants of anthracycline-induced cardiotoxicity was performed, according to results discovered in research on inherited cardiomyopathy disorders. Despite this, we did not detect the same truncating or missense defects of titin previously reported by other groups in our patient [12,13]. The in-frame deletion *TTN*: c.55637_55639delAAG inherited from the patient′s mother is considered to be the most likely pathogenic mutation underlying the patient′s CCM. To our knowledge, this is the first time that a specific *TTN* in-frame deletion (c.55637_55639delAAG) is reported to be associated with anthracycline-related CCM. In contrast, another lymphoma patient with a normal *TTN* missense variant (c.34081C > T) did not have anthracycline-related CCM after treatment with anthracycline-containing regimen.

The molecular mechanisms of CCM are very complex. According to previous studies, the potential causes responsible for CCM might compose of genetic variations relating to metabolism, oxidative stress, inflammation, apoptosis, and autophagy, in addition to structural components in cardiac muscle [20]. We correlate the first-hand clinical findings with the novel *TTN* deleterious mutation and expect that these results can aid in predicting the toxicity of chemotherapy and provide a new view of the research field. Nonetheless, some limitations remain in this study. First, GRCh38 (hg38) is updated from GRCh37 (hg19) to serve as the human reference genome in 2013 [21]. However, due to the foundation referred by the QCII and in-house cohort in the present study, we still used the hg19 reference genome to align the sequencing reads. Second, it is difficult to survey all the possible candidate genes listed in the literature, but our study has included the most CCM-associated 76 genes reported so far. Third, further investigation is needed to confirm the molecular pathway of the newly detected *TTN* in-frame deletion in the CCM pathogenesis. Finally, although we have a significant finding that the specific region deletion (c.55637_55639delAAG) detected in the *TTN* gene profile altered the amino acid sequence (p.E18546del) with a loss-of-function phenotype, more control cases are probably needed to comprehensively demonstrate the pathogenic role of this specific *TTN* variant in chemotherapy-derived CCM.

## 5. Conclusions

Altogether, we are glad to find a likely *TTN* mutation leading to the susceptibility of anthracycline cardiotoxicity in this patient. It is hoped that this finding can contribute valuable information to the field of CCM research, and we plan to conduct more investigations in the near future.

## Figures and Tables

**Figure 1 ijms-23-09261-f001:**
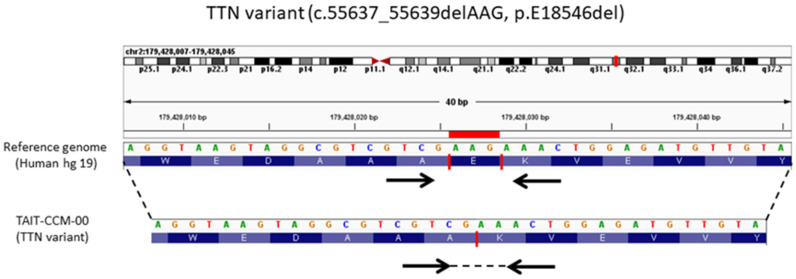
*TTN* in-frame deletion (*TTN*: c.55637_55639delAAG) in the CCM case.

**Table 1 ijms-23-09261-t001:** Targets genes studied in this project.

Gene	Ensembl Gene ID
*ABCB1*	ENSG00000085563
*ABCC9*	ENSG00000069431
*ACTA1*	ENSG00000143632
*ACTN2*	ENSG00000077522
*ANK2*	ENSG00000145362
*ANKRD1*	ENSG00000148677
*ATP1A2*	ENSG00000018625
*BAG3*	ENSG00000151929
*CACNA2D2*	ENSG00000007402
*CASQ2*	ENSG00000118729
*CAT*	ENSG00000121691
*CELF4*	ENSG00000101489
*CORIN*	ENSG00000145244
*CRYAB*	ENSG00000109846
*DES*	ENSG00000175084
*DSP*	ENSG00000096696
*ERBB3*	ENSG00000065361
*FLNC*	ENSG00000128591
*ILK*	ENSG00000166333
*KCNH2*	ENSG00000055118
*KCNQ1*	ENSG00000053918
*LAMA4*	ENSG00000112769
*LDB3*	ENSG00000122367
*MIB2*	ENSG00000197530
*MYH6*	ENSG00000197616
*MYH7*	ENSG00000092054
*MYL10*	ENSG00000106436
*MYL5*	ENSG00000215375
*MYL6B*	ENSG00000196465
*MYL7*	ENSG00000106631
*MYLK*	ENSG00000065534
*MYLK4*	ENSG00000145949
*MYLKP1*	ENSG00000228868
*MYO10*	ENSG00000145555
*MYO16*	ENSG00000041515
*MYO1A*	ENSG00000166866
*MYO1G*	ENSG00000136286
*MYO1H*	ENSG00000174527
*MYO3A*	ENSG00000095777
*MYO3B*	ENSG00000071909
*MYO5BP1*	ENSG00000235130
*MYO5BP2*	ENSG00000238245
*MYO6*	ENSG00000196586
*MYO7A*	ENSG00000137474
*MYO7B*	ENSG00000169994
*MYOF*	ENSG00000138119
*MYOG*	ENSG00000122180
*MYOM2*	ENSG00000036448
*MYOM3*	ENSG00000142661
*MYPN*	ENSG00000138347
*MYRF*	ENSG00000124920
*MYRFL*	ENSG00000166268
*MYSM1*	ENSG00000162601
*MYT1L*	ENSG00000186487
*NEBL*	ENSG00000078114
*NEXN*	ENSG00000162614
*NRAP*	ENSG00000197893
*PKP2*	ENSG00000057294
*PRDM16*	ENSG00000142611
*PRKAG2*	ENSG00000106617
*RAC2*	ENSG00000128340
*RARG*	ENSG00000172819
*RBM20*	ENSG00000203867
*RYR2*	ENSG00000198626
*SCN5A*	ENSG00000183873
*SGCD*	ENSG00000170624
*SLC22A16*	ENSG00000004809
*SLC28A3*	ENSG00000197506
*TGFBI*	ENSG00000120708
*TMEM43*	ENSG00000170876
*TMPO*	ENSG00000120802
*TNN*	ENSG00000120332
*TNNI1*	ENSG00000159173
*TNNT2*	ENSG00000118194
*TNNT3*	ENSG00000130595
*TTN*	ENSG00000155657

**Table 2 ijms-23-09261-t002:** Alterations of cardiac disease in the CCM case.

Gene	Alteration	Function	Impact	Population Frequency (East Asia, gnomAD)	Pathogenicity ^1^
*MYO1A*	c.1630C > T p.R544W	Normal	Missense	0.17%	Uncertain significance
*TTN*	c.55637_55639delAAG p.E18546del	Loss	In-frame deletion	0.05%	Uncertain significance
*RYR2*	c.9336T > C p.I3112I	Normal	Synonymous	0.006%	Likely benign

^1^ QCII interprets the pathogenicity based on the computed points: pathogenic (above 0.98 points); likely pathogenic (0.90 to 0.98 points); variants of unknown significance (VUS, −0.89 to 0.89 points); likely benign (−0.90 to −0.98 points); benign (Below −0.98 points). Determination of pathogenicity referred to the criteria adapted mainly from the 2016 ACMG/AMP guidelines for germline sequence variant interpretation.

**Table 3 ijms-23-09261-t003:** The information of enrolled volunteers in the present study.

Case No.	Sex	Diagnosis	Anthracycline	CCM
TAIT-CCM-00	Male	AML	+	+
TAIT-CCM-01 (CCM parents)	Male	Health donor	_	_
TAIT-CCM-02 (CCM parents)	Female	Health donor	_	_
TAIT-CCM-03 (CCM parents)	Male	Health donor	_	_
TAIT-CCM-04	Female	Health donor	_	_
TAIT-CCM-05	Female	Health donor	_	_
TAIT-CCM-06	Female	Health donor	_	_
TAIT-CCM-07	Male	Health donor	_	_
TAIT-CCM-08 (DNA QC fail)	Male	Colon cancer	_	_
TAIT-CCM-09	Female	Colon cancer	_	_
TAIT-CCM-10	Female	Colon cancer	_	_
TAIT-CCM-11	Male	Colon cancer	_	_
TAIT-CCM-12	Female	Breast cancer	_	_
TAIT-CCM-13	Male	Lymphoma	+	_
TAIT-CCM-14	Female	Colon cancer	_	_

“+” depicts the patient receiving the Anthracycline therapy or displaying CCM syndrome.

**Table 4 ijms-23-09261-t004:** QCII computed results of *TTN* gene.

Case No.	Alteration	Function	Impact	Population Frequency (East Asia, gnomAD)	Pathogenicity
TAIT-CCM-00	c.55637_55639delAAG p.E18546del	Loss	In-frame deletion	0.05%	Uncertain significance
TAIT-CCM-01	No alteration	-	-	_	_
TAIT-CCM-02	c.55637_55639delAAG p.E18546del	Loss	In-frame deletion	0.05%	Uncertain significance
TAIT-CCM-03	c.55637_55639delAAG p.E18546del	Loss	In-frame deletion	0.05%	Uncertain significance
TAIT-CCM-04	No alteration	-	-	_	_
TAIT-CCM-05	c.65504A > G p.N21835S	Loss	Missense	0.44%	Uncertain significance
TAIT-CCM-06	No alteration	-	-	_	_
TAIT-CCM-07	c.13250G > A p.S4417N	Loss	Missense	0.10%	Uncertain significance
	c.23008G > A p.D7670N	Loss	Missense	0.006%	Uncertain significance
TAIT-CCM-09	No alteration	-	-	_	_
TAIT-CCM-10	c.27596G > A p.R9199H	Loss	Missense	0.50%	Likely benign
	c.37143T > C p.A12381A	Normal	Synonymous	0.06%	Likely benign
	c.58211C > G p.S19404C	Loss	Missense	0.54%	Likely benign
TAIT-CCM-11	c.17618T > C p.V5873A	Normal	Missense	0.20%	Uncertain significance
TAIT-CCM-12	c.36157C > T p.R12053W	Loss	Missense	0.15%	Likely benign
TAIT-CCM-13	c.34081C > T p.L11361F	Normal	Missense	0.10%	Likely benign
TAIT-CCM-14	c.1709C > T p.A570V	Normal	Missense	0.74%	Likely benign

## Data Availability

The datasets analyzed in the current study are available upon reasonable request.

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
