# Peer review of "Disclosing an In-Frame Deletion of the Titin Gene as the Possible Predisposing Factor of Anthracycline-Induced Cardiomyopathy: A Case Report"

_ijms, 2022, doi:10.3390/ijms23169261_

Round 1

Reviewer 1 Report

The authors present a very nice case-report, in which they describe the case of 20-year-old patient with acute myeloid leukemia treated with chemotherapy who developed antracycline-induced cardiomyopathy. The whole-exome sequencing was performed and in-frame deletion of the TTN gene was found in this patient that was not previously known to be associated with cardiomyopathy. The same variant was found in his mother and elderly brother. The case is interesting and the report is well-written.

I have one minor comment. I think it is worth to mention which cardiac therapy the patient received. And wether the patient and other cancer patients included as controls received cardioprotectors (for example, dexrazoxane). 

Author Response

Response: Thank you for your comment. It is a valuable suggestion. The relevant content in the manuscript now reads “His left ventricular ejection fraction fell to 11% initially and recovered to 34% after intensive cardiologic care using a combination regimen composing of bisoprolol, valsartan/sacubitril, ivabradine, and spironolactone. Like many other leukemia and cancer patients in Taiwan, he did not take upfront dexrazoxane as a cardioprotective agent along with systemic chemotherapy.” (Line 95-98) and “No cardiomyopathy developed in any of the cancer patients who all had taken systemic chemotherapy without concurrent cardioprotectant dexrazoxane for their diseases.” (Line 146-148)

Reviewer 2 Report

This case seems promising for an opening in this very difficult field. However, the terms "likely benign" and nd "uncertain significancy" leaves much room for doubt. 

Author Response

Response: Thanks for your comments. We have some explanations about the terms of clinical significance based on the guidance for use in ClinVar and QIAGEN Clinical Insight Interpret (QCII) database. To interpret the relationship between the sequence variation and clinical consequence, the American College of Medical Genetics and Genomics (ACMG) has developed the standards and guidance, which recommends the use of specific terms, including “pathogenic”, “likely pathogenic”, “uncertainly significance”, “likely benign”, and “benign”, to describe the gene variants resulted in Mendelian disorders. Both NCBI ClinVar database and QCII analysis software we used in the current study refers the ACMG guidance to reveal those potential pathologic impacts caused by the target gene variants we detected. However, we did specifically interpret the pathogenic phenotypes associate with TTN variants as the specific terms presented in the article so far, which were based on the current information obtained from the QCII database. Of note, the definition of the corresponded terms will be updated in accord with increasing clinical evidence. We will persist to observe the update information from QCII database and renew the consequences of our interesting gene variants. To explain the rationale clearly, we added a short description in the revised version. (Lines 74-78)

Reviewer 3 Report

The study describes a case, of a young adult male diagnosed with AML and upon receiving anthracycline therapy developed severe cardiomyopathy. The study focused on 76 target genes, all of which have been previously associated with chemotherapy-related cardiomyopathy. The authors were able to identify a deletion in the TTN gene that most likely contributes to the development of the patient’s condition. The patient inherited the variant from his mother, which means it is not a result of the genomic changes in his cancer, furthermore, it is present in low frequencies in the population so it is more likely to have pathogenic effects.

Minor comments:

-    table 4: the „gene” column is supposed to be called „case”, the names refer to case numbers. Please correct.

-    Figure 1 legend: aligned to hg19 reference genome, not reference gene. Please correct.

-    There are several grammatical errors in the text which make it difficult to understand it sometimes. Please have someone read through it to correct these mistakes.

Major comments:

-    The hg19 genome assembly is outdated, why not use the hg38 assembly?

-    The control cases do not include AML patients who received anthracycline treatment, let alone developed cardiomyopathy. If possible, include more control cases to illustrate the findings.

Author Response

Minor comments:

 table 4: the „gene” column is supposed to be called „case”, the names refer to case numbers. Please correct.

Response: Thanks for the comment. We have revised the typo.

Figure 1 legend: aligned to hg19 reference genome, not reference gene. Please correct.

Response: Thanks for the comment. We have revised the typo.

There are several grammatical errors in the text which make it difficult to understand it sometimes. Please have someone read through it to correct these mistakes.

Response: Thanks for the suggestion. We have revised some ambiguous sentences and tried our best to correct the grammatical errors in the manuscript.

Major comments:

 The hg19 genome assembly is outdated, why not use the hg38 assembly?

Response: We sincerely appreciate this valuable comment. We have known GRCh37 (hg19) was released in 2009 to be widely used as human reference genome assemble, and the updated version “GRCh38 (hg38)” was published in 2013. To our knowledge, the updated hg38 can improve the locus-specific issues, genome-wide alignments, and false variant callings. However, we do focus our attention on some limitations that still exist in hg38, leading hg19 to remain the reference genome we used in the present study. The first one is an erroneous read alignment issue. Tongqiu Jia’s group has demonstrated that thousands of variant calls are absent, with 641 genes presenting no variation, which is due to the erroneous alignment to the hg38 reference genome (1). Second, comparing to hg38, hg19 reference genome has been used to be the database for cataloguing human gene variants and mapping functional genotype sequences in several large projects and studies to date. In our study approach, sequencing reads were aligned with hg19 human reference genome with potential variants and gene copy number alterations called by using QCII software, which is developed upon hg19 reference genome. Thus, to be concordant with the reference genome database between sequence reading and gene mutation calling, we used hg19 to be the human reference genome in the present study. For interpreting the reason of using hg19, we added the depiction in the discussion section. (Line 195-198)

The control cases do not include AML patients who received anthracycline treatment, let alone developed cardiomyopathy. If possible, include more control cases to illustrate the findings.

Response: Thanks for the kind response. In the present data, we recruited 15 volunteers to observe the gene variations with or without chemotherapy, including 1 AML patient with anthracycline treatment, 7 health donors, 5 colon cancer without anthracycline treatment, 1 breast cancer without anthracycline treatment, and 1 lymphoma with anthracycline treatment. As the results we showed, anthracycline therapy only caused cardiomyopathy in the AML patient with TTN variant, not in the lymphoma patient with wild-type TTN gene. Of note, the TTN variant that occurred in other malignancy patients revealed low CCM risk after receiving non-anthracycline therapy. We hope the present evidence can support that the specific TTN in-frame deletion mutation obtained by the AML case could lead to cardiomyopathy after receiving anthracycline therapy. This finding is consistent with previous reports that defined the TTN variant-derived CCM pathogenesis mechanism. Although the specific region deletion (c.55637_55639delAAG) occurring in the TTN gene profile eventually altered the amino acid sequence (p.E18546del) with a loss-of-function phenotype, we agree with you there is a limitation in this study that more control cases are probably needed to comprehensively demonstrate the pathogenic role of this specific TTN variant in AML chemotherapy-derived CCM (see description in the discussion section, lines 202-206).

Reference:

  1.   Jia T, Munson B, Lango Allen H, Ideker T, Majithia AR. Thousands of missing variants in the UK Biobank are recoverable by genome realignment. Annals of Human Genetics. 2020 May;84(3):214–20.

Round 2

Reviewer 2 Report

The changes included are sufficient